# Evaluation of Parameters Which Influence Voluntary Ingestion of Supplements in Rats

**DOI:** 10.3390/ani13111827

**Published:** 2023-05-31

**Authors:** Santiago Ruvira, Pilar Rodríguez-Rodríguez, Silvia Cañas, David Ramiro-Cortijo, Yolanda Aguilera, David Muñoz-Valverde, Silvia M. Arribas

**Affiliations:** 1Department of Physiology, Faculty of Medicine, Universidad Autónoma de Madrid, C/Arzobispo Morcillo 2, 28029 Madrid, Spain; 2Food, Oxidative Stress and Cardiovascular Health (FOSCH) Research Group, Universidad Autónoma de Madrid, Ciudad Universitaria de Cantoblanco, 28049 Madrid, Spain; 3Department of Agricultural Chemistry and Food Science, Faculty of Science, Universidad Autónoma de Madrid, C/Francisco Tomás y Valiente 7, 28049 Madrid, Spain; 4Institute of Food Science Research (CIAL), Universidad Autónoma de Madrid (UAM-CSIC), C/Nicolás Cabrera 9, 28049 Madrid, Spain; 5Animal House Facility, Veterinary Unit, Faculty of Medicine, Universidad Autónoma de Madrid, C/Arzobispo Morcillo 2, 28029 Madrid, Spain

**Keywords:** food supplement, gelatin, voluntary ingestion, habituation, rats, sex, refinement

## Abstract

**Simple Summary:**

Preclinical studies evaluating the safety and efficacy of new drugs require experimental animals, which is important to ensure both adequate dosage and animal welfare. Administration through voluntary ingestion can improve animal wellbeing. We aimed to develop a protocol using small gelatin cubes to train rats, assessing the influence of age, sex, fasting, flavors (vanilla), and sweeteners (sucralose) to accept the new food. We tested the usefulness of the protocol to supplement rats during lactation. We demonstrated that most animals were easily trained to accept the gelatin cube in 2–3 days. However, some rats refused the new food even after 8 days. The proportion of rats that did not train was higher in adult males compared to young males, adults, and young females, suggesting the influence of sex. Four-hour food deprivation reduced the time for acceptance only in females, but flavoring or sweeteners in the gelatin did not modify it. Rats trained prior to gestation remembered training 2 months later and ate a gelatin containing a supplement daily during lactation for 1–5 min, without problems with the pups. We conclude that gelatin-based supplementation can be used for drug studies in rats, ensuring adequate dosage and wellbeing, which is important for the detection of non-trained rats.

**Abstract:**

Drug safety and efficacy studies frequently use oral gavage, but repetitive usage may cause problems. Administration through voluntary ingestion represents an opportunity for refinement. We aimed to develop a protocol for voluntary ingestion of gelatin-based supplements in rats, assessing the influence of age, sex, fasting (4 h), and additives (vanilla, VF; sucralose, S), and to test it in lactating dams. Three-week-old and 5-month-old Sprague-Dawley rats were placed individually in an empty cage containing a gelatin cube and trained daily (5 days/week), recording the day the whole cube was consumed (latency). Rats trained prior to gestation were offered a gelatin containing 250 mg/kg cocoa shell extract (CSE) during lactation. Rats that did not eat the cube after 8 training days were considered non-habituated, with a proportion similar in young males (7.1%), young females (11.1%), and adult females (10.3%), but significantly higher in adult males (39.3%). Excluding non-habituated rats, latency was 2–3 days, without differences between young and adult rats (*p* = 0.657) or between males and females (*p* = 0.189). VF or VF + S in the gelatin did not modify latency, while fasting significantly reduced it in females (*p* = 0.007) but not in males (*p* = 0.501). During lactation, trained females ate the CSE-gelatin within 1–5 min without litter problems. Conclusions: Acceptance of a gelatin-based supplement is negatively influenced by male sex, facilitated by fasting, and not modified by additives. Training is remembered after 2 months and does not interfere with lactation. Gelatin-based voluntary ingestion is suitable to administer drugs that need to pass through the digestive system, ensuring adequate dosage, and is important to detect non-habituated rats prior to the study. The current protocol may be implemented by training the rats in their own cage.

## 1. Introduction

Animal experimentation constitutes a relevant aspect of biomedical research and a key piece of preclinical research to evaluate the safety and efficacy of new drugs. This type of research requires repetitive administration of substances, which is a critical component of experimental design and represents a good opportunity for refinement [1].

The development of nutraceuticals or the testing of drugs that need to pass through the digestive system involves the oral administration of selected compounds. The most common form is oral gavage, which consists of force-feeding a substance with a tube inserted in the mouth and then into the stomach. Gavage provides a rapid and efficient means of accurately delivering oral dosing to rodents, and it is applicable to unanesthetized animals. However, oral administration is not exempt from serious complications, including reflux, aspiration, esophageal irritation, and inflammation. The main complication is reflux and respiratory aspiration, which have been reported to cause around 20% mortality in some studies, depending on dosage [2,3]. Besides, the procedure causes stress, as evidenced by the increase in corticosterone levels [4,5]. Even though these alterations may be minimized by appropriate handling by an experienced technician [6], they may interfere with animal welfare and with the interpretation of results. Thus, reducing the stress associated with chronic drug delivery to experimental animals is desirable. Besides, the use of oral gavage may be particularly detrimental in periods such as gestation or lactation since it may influence maternal behavior with detrimental effects on pups [7].

An alternative to oral gavage is the administration of the compound of interest mixed with an attractive vehicle by voluntary ingestion. Among vehicles, jam [8], gelatin [9], cookie dough [10], or nut paste [11] have been successfully used in rodents. One of the challenges is to guarantee adequate dosage, i.e., to ensure full ingestion, particularly if the substance under study is bitter [10], and most of the above-mentioned studies include flavoring substances and sweeteners in the vehicle. It must be noted that for some types of studies, it would be preferable to avoid a vehicle with a high caloric content. 

In the voluntary acceptance of supplements, a key issue is the fact that rodents have a natural neophobia, and recent data indicate that this may be modulated by sex and age [12]. Therefore, our objective was to evaluate several factors that may influence the voluntary ingestion of a novel food in rats and refine protocols for supplement administration. Using gelatin cubes (GC), we studied in Sprague-Dawley rats the effect of intrinsic (sex and age) and extrinsic (additives and fasting) factors on (1) the percentage of successfully trained rats, (2) the training time requested for full acceptance (latency to ingestion), and (3) the capacity to recall after a period of training. Besides, we have also evaluated the effectiveness of the training to supplement lactating rats with an extract derived from cocoa shell (CSE), a by-product derived from the chocolate manufacturing industry rich in bioactive compounds with potential to reduce cardiometabolic diseases, which we have previously analyzed in vitro [13]. Based on our findings, we propose some key points that may be useful for researchers willing to use this method of supplementation.

## 2. Materials and Methods

### 2.1. Experimental Animals

Sprague-Dawley rats from the colony bread at the animal house facility of the Universidad Autónoma de Madrid (ES-28079-0000097) were used at the following age points: 3-week-old (young rats; *n* = 9 females and *n* = 14 males); 5-month-old (adult rats; *n* = 35 females and *n* = 40 males). We also used 7 lactating rats to evaluate the acceptance of a gelatin containing a compound of interest (CSE) after training. The experimental procedures conformed to the Guidelines for the Care and Use of Laboratory Animals (National Institutes of Health publication no. 85-23, revised in 1996), the Spanish legislation (RD 53/2013), and the Directive 2010/63/EU on the protection of animals, and were approved by the Ethics Review Board of Universidad Autónoma de Madrid and the Regional Committee of Comunidad Autónoma de Madrid (PROEX 19/04; approval date: 20 March 2019). 

After weaning, rats from the same sex were housed in groups of 3–5 per box and maintained in type III cages (24 × 19 × 45 cm; length × height × width) or type IV cages (55 × 18 × 32 cm; length × height × width), according to the number and weight of the rats, with poplar bedding. Cellulose nestlets and play tunnels (Index Research S.L., Madrid, Spain) were used for environmental enrichment. The animals were constantly kept under controlled conditions of temperature (22 °C), humidity (40%), and photoperiod of 12 h of darkness and 12 h of light. Rats were fed *ad libitum* (except during fasting periods) with a diet containing 51.7% carbohydrates, 21.4% protein, 5.1% lipids, 3.9% fiber, 5.7% minerals, and 12.2% humidity (SafeA03; Safe Augy, France). Drinking water was also provided *ad libitum*. The animal health was regularly monitored by staff, ensuring rats were free from pathogens that may interact with any of the parameters studied.

### 2.2. Gelatin Cubes Preparation

The cubes for training were prepared with 100% bovine gelatin (Inkafoods, S.L., Barcelona, Spain) in water at a concentration of 140 g/L. Water was first heated at 50–60 °C in a glass beaker, and the gelatin was slowly added, stirring it until complete dissolution. At this point, the different additives can be incorporated into the mixture: vanilla flavor (VF, 4.8 mL/L; MyProtein, Hut.com Ltd., Manchester, England) as a non-caloric flavoring agent, alone or with sucralose (S, 0.6 g/L; sucralin, sucralose S.L., Barcelona, Spain). The mixture was transferred to a mold, ensuring homogeneous distribution, to prepare 1 cm^3^-sized GC, since preliminary results showed that this size is easily handled by rats (Appendix A). The mold was left to cool down to allow solidification, first at room temperature and then in the fridge. Thereafter, the individual cubes were extracted from the mold and stored in the fridge in plastic bags, or, if the gelatin was not going to be consumed immediately, they were stored frozen at −20 °C. Cubes can be defrosted at room temperature without changing their consistency or shape. 

**Cubes with cocoa shell extract (CSE).** Cocoa shell was kindly supplied by Chocolates Santocildes S.A. (https://www.chocolatessantocildes.com/ (accessed on 10 March 2023), Castilla y León, León, Spain). An extract (CSE) rich in phenolic compounds was prepared as previously described [14]. The CSE was incorporated after the gelatin was dissolved. The dose of CSE used for the study was 250 mg/kg. The animals were first weighed, and the cubes were prepared according to the weight of the rat for supplementation during the entire lactation period.

### 2.3. Administration Protocols

**Training protocol.** The rat was placed in an individual empty box without bedding with the GC and left for 2 h. At the end of this period, we recorded if the animal ingested part or the whole cube. This procedure was carried out for 5 days/week (Monday to Friday) at the same time (9:00 to 11:59 a.m.). The first day that the rat ingested the entire GC was considered the acceptance day, and the number of days from the first exposure was counted as the latency period. The cubes that were not eaten were discarded.

**Fasting protocol.** To study the influence of fasting, a group of adult males and females were deprived of feed in their own cage for 4 h prior to the training protocol (from 7:00 to 11:00 a.m.).

**Training protocol in their own cage.** In a group of rats who did not accept the GC in an empty cage, the cube was presented individually in the usual cage, placing it in the hopper without food.

**Recall protocol.** A group of adult males and females previously trained were exposed to the NF gelatin 1 month later, and the latency to ingestion was recorded. In these rats, the time needed for complete ingestion was also registered.

**Protocol for CSE supplementation in lactating rats.** Seven female rats previously trained were mated, and after giving birth, they were weighed to prepare the CSE GC at a dose of 250 mg/kg. From the second day postpartum, the rats were offered the CSE cube 5 days/week during the entire lactation period. For this protocol, once the rat ate the whole cube, it was immediately returned to the cage with the pups. 

### 2.4. Statistical Analysis

Data analysis was performed by GraphPad software (GraphPad Prism, version 8.0, Boston, MA, USA). The distribution of the variables was evaluated using the Kolmogorov–Smirnov test. Since some of the data did not follow a normal distribution and the sample size was small in some protocols, quantitative data were reported as the median and maximum and minimum levels, and statistical analysis was performed by Kruskal–Wallis or Mann–Whitney’s U tests. To assess differences in the proportion of non-habituated rats, Fisher’s exact test was performed. Significance was established with a *p*-value (*p*) < 0.05. 

## 3. Results

We evidenced that some rats did not eat the gelatin after 8 days of training. Some preliminary data indicated that they did not eat it even if exposed for 3 weeks. Based on this, and to avoid unnecessary stress, rats that did not eat the gelatin cube in 8 days were counted as “non-habituated”. 

### 3.1. Influence of Age and Sex on Habituation and Latency

In adult males, the proportion of non-habituated rats was significantly higher compared to young males (χ^2^ = 4.728; *p* = 0.03), while no significant difference was detected between young and adult females (χ^2^ = 0.004; *p* = 0.948). In young rats, the proportion of non-habituated rats was similar between sexes (χ^2^ = 0.109; *p* = 0.742), while in adult rats it was higher in males compared to females (χ^2^ = 6.440; *p* = 0.011) (Figure 1A). We evaluated differences in the latency to ingestion in all rats, considering 8 days of latency for non-habituated rats. Latency was significantly higher in adult males compared to young males and of near statistical significance in adult males compared to adult females (*p* = 0.056). No significant differences were found between young and adult females (Figure 1B).

We assessed the latency to ingestion in the population of rats that accepted the novel food (i.e., excluding non-habituated rats). We did not find statistical differences between groups, suggesting no influence of sex or age on latency (Figure 2). 

In a group of 5 non-habituated males, we evaluated if the presentation of the GC in their own cage would improve acceptance. We observed that they did not try the GC when placed in the empty cage. However, on the same day, all tried the GC if it was placed in the food hopper of their own cage (Appendix A). The following day, 4 out of the 5 rats ate the cube when the gelatin was placed in the empty cage. 

### 3.2. Influence of Additives and Fasting on Habituation and Latency

The influence of additives (NF, VF alone, or VF + S) was evaluated in adult rats. Additives did not influence the proportion of non-habituated rats, which was similar in both males (χ^2^ = 1.504; *p* = 0.471) and females (χ^2^ = 0.932; *p* = 0.627) (Figure 3A). Similarly, the latency to ingestion, including all rats, did not show significant differences between groups (NF, VF, and VF + S) either in males or females (Figure 3B). 

Latency to ingestion, excluding non-habituated rats, was not statistically different between groups (NF, VF, or VF + S) in males or females (Figure 4).

In a group of adult rats trained with NF, we evaluated the time needed for complete ingestion. Female rats were faster (mean time = 1.7 ± 0.2 min, minimum = 1.25 min, and maximum = 2.1 min; *n* = 4) compared to males (mean time = 7.9 ± 1.5, minimum = 4.0 min, and maximum = 11.5 min; *n* = 4; *p* = 0.02).

We also analyzed if short (4 h) fasting affected the latency to ingestion. We observed that 4 h prior fasting did not modify latency in males (Figure 5) but fasting reduced it significantly in females (Figure 5).

### 3.3. Capacity to Remember Training and Acceptance of GC with a Compound of Interest

In a group of 6 adult males and 6 adult females previously trained with NF gelatin, we evidenced that after 1 month, all rats, except one female, ate the GC. 

We also tested the capacity to accept a supplement (CSE) incorporated into the GC in a group of 7 previously trained adult females during the lactation period. All the dams accepted the gelatin and ate it during the 3 weeks of lactation and completed ingestion in a single attempt and between 1 and 5 min. No alterations in dam behavior or problems with the pups were detected, compared to our previous studies in lactating rats [15]. A video showing the ingestion of a CSE cube by a lactating rat is included (Appendix A).

## 4. Discussion

Experimental animals are an essential tool for the development of new drugs; they are commonly used to evaluate efficacy and toxicity. The procedures involve repetitive administration of the compound of interest, ensuring adequate dosage. Some drugs can be injected, while others, such as nutraceuticals, need to pass through the digestive tract to exert their actions. Ingestion is also a method required to evaluate drug activity for the development of orally administered medicines. For these studies, voluntary ingestion methods have several advantages over involuntary procedures (such as oral gavage), the most important being that they are non-invasive, reducing stress and possible complications [2,3,4,5], particularly during sensitive periods such as gestation and lactation. One possible way of delivery is to include the substance of interest in the drinking water. Although this method is suitable and easy, some substances cannot be incorporated, i.e., if they are hydrophobic, and daily dosage may be subject to differences in the amount of water each animal drinks. The present study was conducted to implement a protocol for voluntary ingestion of food supplements in rats based on a gelatin matrix, evaluate key factors influencing acceptance, and assess the capacity to remember previous training. Once the training protocol was established, we also evaluated its effectiveness as a supplement with a compound of interest (CSE) during lactation, a critical period when oral gavage should be avoided. Our main results are that training rats for voluntary ingestion of a GC is easy since it requires very basic equipment and does not require specific skills, and it is a suitable method for supplementation during lactation. Regarding factors that affect training, we detected a higher proportion of adult males who did not habituate, although acceptance may be improved if the training is conducted in a familiar environment. Based on our findings, we suggest that it is important to detect animals who refuse to eat the gelatin during training to exclude them from the study, avoiding unnecessary stress and experimental bias. These animals may be useful for other studies that do not require voluntary ingestion of compounds, contributing to the three Rs. In Appendix B, we propose a protocol for voluntary ingestion of supplements in rats. 

There are several vehicles to incorporate the compounds of interest for voluntary supplementation in rodents (jams, cookies, paste, and gelatins). We chose gelatin as a vehicle since it is cheap and easy to produce and because we could incorporate all types of compounds into it. Besides, it can be shaped in various sizes, and the rats can handle it easily without breaking into pieces (Appendix A), as previously described in mice [16]. Thus, this vehicle ensures complete ingestion and adequate dosage. Besides, gelatin is a non-caloric vehicle and therefore suitable for studies analyzing compounds interfering with metabolic processes. In fact, it has been proposed that it can be used to deliver glucose for tolerance tests as a substitute for oral gavage or injection since it causes less stress and guarantees a better interpretation of results.

Caution is a natural response of animals to unfamiliar objects, and the term “neophobia” is used to characterize fear-like responses based on the novelty of a stimulus, regardless of its modality. Neophobic responses may occur to objects, places, or sounds, and food neophobia applies to the refusal or suppression of the ingestion of a new food. Rodents are neophobic animals, with different degrees depending on the strain of mice or rats and if they are wild or laboratory-bred [17]. When a rat is exposed to a new food, the initial response is avoidance, followed by gradual sampling, and, if there is no harm, consumption increases across successive encounters [17,18]. Our experiments showed that Sprague-Dawley rats took a median of 2–3 days to accept a novel food (a GC), a latency like that reported in mice (ranging from 2–4 days) [19] or in Lister-hooded rats, who needed 3 days of training for full acceptance [20]. 

It is important to note that we detected a percentage of rats that maintained neophobia over a period of 2 weeks, and we considered these rats non-habituated. The proportion of non-habituated rats was higher in adult males compared to the other groups. Maintained neophobia has been previously detected in male mice, where it was shown that 5% refused to eat a gelatin even after prolonged fasting [16,19], and in Long Evans rats, where it was reported that 4 out of 8 males never ate the novel food [21]. We observed that the proportion of non-habituated rats was larger in adults but not in young males, which, according to our previous data, are still in the prepubertal stage [22]. Therefore, our data indicate an influence of sex and age on the degree of neophobia, with a higher susceptibility among adult males. These difficulties in habituation can be related to a previous high level of stress that drives food avoidance [17,21], for example, due to laboratory noise [16]. Since gonadal hormones play a key role in the regulation of the hypothalamic–pituitary–adrenal axis [23], it is possible that male rats are more susceptible to maintaining neophobia in response to environmental stressors. 

In non-neophobic rats, we also analyzed if sex and age affected time to acceptance, i.e., latency. We did not find sex differences, in contrast to the findings of other authors, who reported that Long Evans male rats habituated to a novel food faster than females if exposed to an unfamiliar environment, although no significant differences between sexes were found if the new food was presented in their own cage [21]. This is likely related to the fact that rats respond with neophobia not only to the new object but also to the container [17]. The faster adjustment of males to an unfamiliar environment has been explained by their higher risk-taking behavior compared to females [24]. We did not find significant differences in latency between young and adult rats. Some studies evidence a better acceptance of a new food in adolescent Sprague-Dawley rats, showing that they are more sensitive than adults to the hedonic properties of an appetitive stimulus and less sensitive to aversive stimuli, such as quinine [25]. Even though we did not detect differences in latency, the fact that young males had a significantly lower rate of non-habituation compared to adults suggests the interest in training rats at a young age for supplementation studies. We did not use old rats, but it is possible that they would have a higher latency since neophobia habituation shares neural circuits with mechanisms of declarative memory [18], and studies comparing adult and old rats show increased neophobia in aging together with deficits in spatial learning [26]. 

Regarding the time the rats need for GC intake once they are trained, our data are in accordance with other studies showing that rodents take an average of 1 to 10 min for complete ingestion. Zhang and co-workers using a gelatin-based supplement reported that most mice ate it within 1 min of presentation and finished the entire piece in a single attempt [19]. Neto et al. using female mice trained with a cookie reported that the animals need 5–10 min [27], and Lister hooded rats can achieve a complete intake within 5 min [20]. We evidenced that male rats took a longer time for complete ingestion, which is in accordance with data from several rat strains showing that females eat faster and approach the food before feeding more frequently than males [17]. However, the study of Teixera-Santos and co-workers in mice showed a lower latency in males, who were able to eat the supplement in 1 min compared to females, who took 5 min [8]. Thus, species differences may occur. 

We also analyzed the capacity of external factors (additives and prior fasting), which may facilitate training and can be incorporated into future protocols. Rodents refuse some substances, such as alcohol, due to aversive odors, as well as foods with a bitter or acidic taste. It has been demonstrated that acceptance can be improved if the supplement includes a flavoring agent or a sweetener. In laboratory mice, vanilla flavor increases nicotine consumption [28]. House mice preferred foods with flavors (vanilla, chocolate, hazelnut, or peanut) to natural foods, while rats rejected chocolate flavor and did not show a preference for the other additives [29]. Rats also increase alcohol consumption in the presence of sweeteners (sucralose, saccharine, or maltodextrins) [30,31], and sugar-free fruit juices improve the acceptance of nicotine [20]. In our study, we did not observe a reduction in latency or in the proportion of non-habituated rats using gelatins with vanilla with or without sucralose; we even found worse results than with neutral gelatin. This is in accordance with the study of Sclafani and Clare using the same strain of rats, showing that sucralose reduced food palatability in Sprague-Dawley rats [32]. We conclude that the food additives and sweeteners tested (vanilla and sucralose) do not improve the acceptance of gelatin in Sprague-Dawley rats.

We also tested the influence of prior fasting since it has been shown that the hungrier the rat, the quicker it starts to eat an unfamiliar food [33]. This was confirmed in our study in females, which demonstrated that prior food deprivation reduced latency. However, this was not observed in males. We used a short fasting period of 4 h compared to previous studies using overnight fasting [19], and it would be possible to reduce it since it has been shown that 30 min fasting is sufficient for the acceptance of a jam including a drug in mice [8]. Although it has been demonstrated that fasting for less than 16 h does not represent a stress for the rat [34], and we demonstrated a positive effect on latency, reducing acceptance by 1 day, we think that it is not strictly necessary for the training protocol. 

Finally, one of our aims was to evaluate the effectiveness of the training protocol for supplementation in rats during lactation, which, together with gestation, is a very sensitive period where stress induced by oral gavage should be avoided. In fact, stress induced by oral gavage in the rat leads to weight loss, and although this can be minimized if the rat is anesthetized [35], this can be a sign of stress with a negative influence, particularly during gestation and lactation. Rats subjected to prenatal stress around the gestational period have more spontaneous abortions and fewer viable pups [36], and oral gavage during pregnancy alters the behavioral development of the offspring [37]. During lactation, dams exposed to chronic social stress show impaired maternal care and lactation [38]. We demonstrated that 100% of trained female rats prior to gestation remembered the training two months later and ate the gelatin, including the supplement (CSE), on a single attempt in 1–2 min. This procedure greatly reduces the rat’s stress since the animal is returned to the cage once the gelatin has been eaten, and we have evidenced no modifications in the dam or pups’ behavior. Besides, complete gelation consumption ensured an adequate dosage of the compound of interest. We also have evidence from a metabolomic study that CSE bioactive compounds (caffeine and theobromine) are present in rat plasma after 1 week of supplementation [13]. Therefore, this training protocol is suitable for experiments during gestation and lactation. 

**Limitations and protocol implementation.** The present study aimed to improve methods of supplementation by voluntary ingestion by assessing some factors that may affect neophobia and latency. We evidence the feasibility of training rats to eat a gelatin-based vehicle and the capacity to remember and accept the supplement months later. The protocol is an alternative to oral gavage and improves animal wellbeing. However, it is not devoid of limitations, the main one being the resistance of some rats, mainly adult males, to training; therefore, a pre-screening of animals is required. We also showed that males do not improve latency by short fasting. As indicated above, males may have more neophobia due to the higher influence of stress factors. One of them is isolation in an empty cage, which was used to facilitate the view of the gelatin and avoid hiding or breaking it. However, this is an unfamiliar environment, which may contribute to neophobia [17]. Besides, we exposed the rats to repeated short isolation periods, which have been shown to also be a stressor [39]. Presentation of the novel food in their cage may facilitate acceptance and has been previously used in rats [10] and mice [19], placing the supplement in a glass Petri dish overnight. However, this protocol requires isolating the rat for the whole night. We used an alternative method in the group of adult males who refused to eat the cube with the usual protocol in the empty cage after isolation and observed that if the cube was placed in the hopper, where they usually receive the food, they accepted it. Moreover, after this test, most rats ate the gelatin with the usual protocol in the empty cage in the following days. It has also been shown that acceptance of a novel food does not depend solely on the individual’s experience but has a social component. For example, if a mouse has tried a novel food, it will be investigated by the other mice in the cage, which may result in other animals from the colony selecting food with the same scent [40]. We also observed this behavior when presenting the gelatin in the hopper, and it is possible that an initial training in the regular cage may also help acceptance through this social process. However, after training, the supplement should be given individually to ensure dosage. Therefore, we suggest implementing our protocol by training rats with the gelatin in their usual environment, followed by individualization in the empty cage to ensure dosage.

Neophobia has an important genetic component, and studies in KO mice have identified some genes influencing it. Thy-1 (a cell adhesion molecule) KO mice display neophobia, which has been suggested to be mediated by neuronal plasticity regulation [40]. In addition, a study of wild brown rats living in various Tokyo locations demonstrates differences in neophobia and the existence of rats that are indifferent to novel objects, probably due to genetic variations [41]. These data suggest the possibility of obtaining genetically modified animals with lower neophobia through selective breeding.

Regarding the utility of this method, we think it is not restricted to food supplements, but it would be a way of testing the effectiveness of drugs. Some studies using voluntary administration of analgesics with gelatin-based vehicles have found poorer results compared to intraperitoneal administration [42], likely due to modifications of the drug by passage through the gastrointestinal tract. However, it has to be noted that this study evaluated analgesia after a short period (1 h), and it is possible that long-term studies would yield better results. In addition, oral intake of drugs is the preferred route of administration, and voluntary supplementation would be a useful method for this purpose. 

## 5. Conclusions

The present study demonstrates the feasibility of training rats to voluntarily accept a supplement based on a gelatin vehicle that can be used during lactation and gestation. Training is not influenced by additives, but it is improved by fasting in females. Male sex may negatively influence neophobia, making it important to detect non-habituated animals to exclude them from the study and avoid unnecessary stress. It is possible that this neophobia can be reduced by the initial presentation of the novel food in their own cage. We propose some tips for future use of this protocol in Appendix B.

## Figures and Tables

**Figure 1 animals-13-01827-f001:**
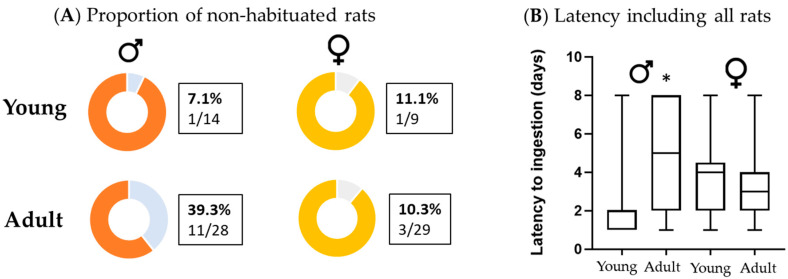
Influence of sex and age on the proportion of non-habituated Sprague-Dawley rats (**A**) and latency to ingestion including all rats (**B**). Young (3-week-old) and adult (5-month-old) rats. Data in (**B**) show the median and maximum and minimum days of latency. Number of rats from each group is shown in (**A**). Statistical analysis was performed by Fisher’s exact test (**A**) or Kruskal–Wallis test (**B**); * *p*-value < 0.05 adult males compared to young males.

**Figure 2 animals-13-01827-f002:**
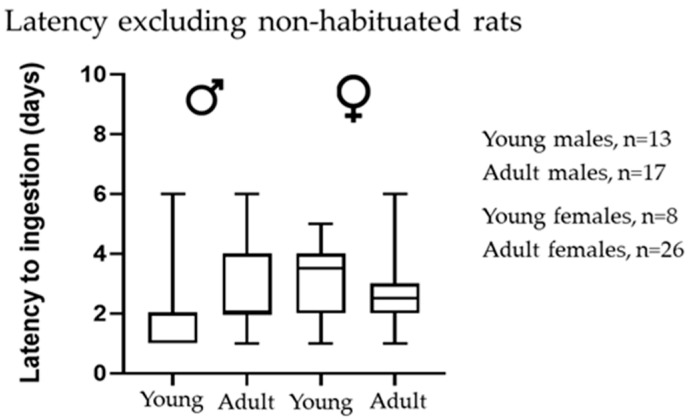
Influence of sex and age on the latency to ingestion excluding non-habituated rats. Young (3-week-old) and adult (5-month-old) rats. Data show the median and maximum and minimum days of latency. Statistical analysis was performed by Kruskal–Wallis test; *n*, number of rats.

**Figure 3 animals-13-01827-f003:**
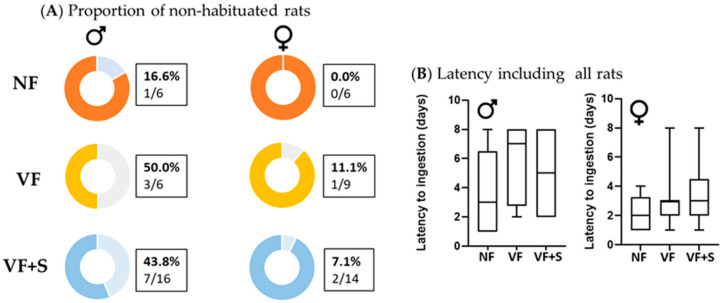
Influence of additives on the proportion of non-habituated Sprague-Dawley rats (**A**) and latency to ingestion excluding non-habituated rats (**B**). Young (3-week-old) and adult (5-month-old) rats. NF, neutral flavor; VF, vanilla flavor; VF + S, vanilla flavor, and sucralose. Data in (**B**) show the median and maximum and minimum levels of latency. Number of rats from each group is shown in figure (**A**). Statistical analysis was performed by Fisher’s exact test (**A**) or Kruskal–Wallis test (**B**).

**Figure 4 animals-13-01827-f004:**
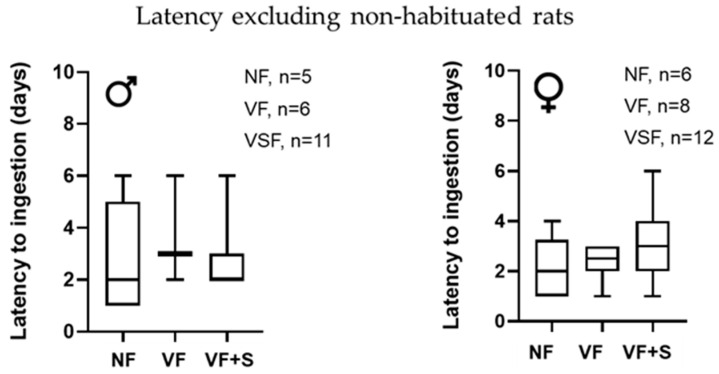
Influence of additives on latency to ingestion excluding non-habituated rats. Young (3-week-old) and adult (5-month-old) rats. NF, neutral flavor; VF, vanilla flavor; VF + S, vanilla flavor, and sucralose. Data show the median and maximum and minimum days of latency. Statistical analysis was performed by Kruskal–Wallis test; *n*, number of rats.

**Figure 5 animals-13-01827-f005:**
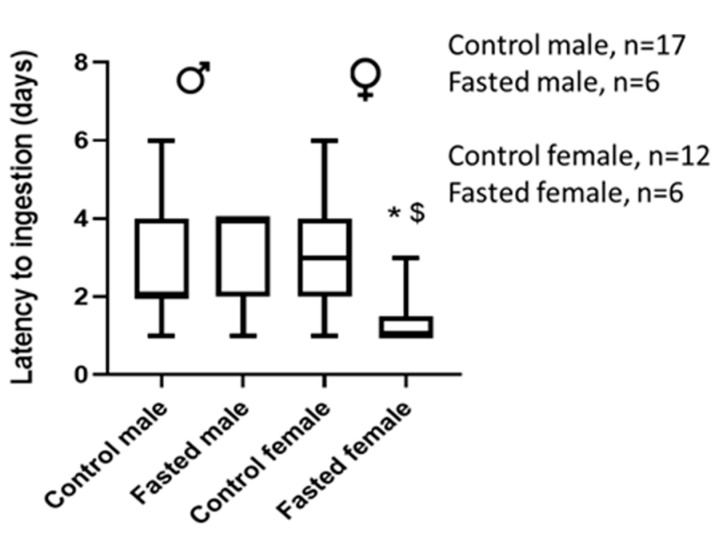
Effect of 4 fasting on latency to ingestion in adult male and female rats excluding non-habituated rats. Data show the median and maximum and minimum days of latency. Statistical analysis was performed by Mann–Whitney’s U test; * *p*-value < 0.05 with control females, $, *p*-value < 0.05 with fasted adult males; *n*, number of rats.

## Data Availability

The data are available upon request to the corresponding author by institutional email and after ethical evaluation.

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
