# Peer review of "Evaluation of Parameters Which Influence Voluntary Ingestion of Supplements in Rats"

_animals, 2023, doi:10.3390/ani13111827_

Round 1

Reviewer 1 Report

Discuss possible genetic selection or reproductive selection to increase proportion of rats habituated to the consumption of gelatin. Create a lineage that consumes voluntarily easily.

P7L275 It says “dugs” instead of “drugs”  

Please indicate the novelty, relevance or originality of the study

Author Response

ANSWER. We would like to thank you for the suggestions and comments, which we have incorporated in the new version.

We have now included in the discussion the issue of the possible selection of non-neophobic rats in a new section at the end of the discussion “limitations and protocol implementation” (lines 429-435).

Regarding the issue of novelty and relevance of our study, we agree on the fact that neophobia has been well studied previously. We studied this aspect with a different perspective, since our aim was not to gain insight in this process, but to explore factors affecting voluntary ingestion of supplements to provide key elements to develop a protocol to substitute oral gavage by voluntary ingestion. The novelty of our study is that we approach the study of neophobia from the practical point of view. The results from our study are relevant from the animal welfare point of view, since we provide sufficient information for researchers to use this method of supplementation which is less aggressive. We have also corrected the typo.

Reviewer 2 Report

Dear Authors,

thanks for the nice study and the nice idea to refine oral gavage in rats. Oral gavage is stress for rats and humans and a method to circumvent this situation is highly appreciated from my side! So, a method to use voluntary eating would be perfect. Unfortunately, not all animals respond as they should, which is always a problem using voluntary appetitive behavior.

I have one main question and some points to clarify, so this would be a minor revision.

My main question is: The results part seems to be three experiments, one with the question “response to training in the focus of age and sex”, the second with the question of the flavor and third the lactating issue. First and third, the animal numbers are clear, but the second experiment, I get confused. Are the animals used again (so they already had undergone the training from part one)? That would mean they are not naïve anymore which would perhaps give an idea why they are more “non-habituating” to the flavors? If this is a new group, then the animal numbers in the method part are not fitting.

Perhaps you can clarify my confusion.

Then some less important questions:

·        Have the rats/animal housing facility SPF status

·        Did you change the cage for fasting the rats? As rats eat when fed ad lib. not very “clean” but rather with a lot of crumbles, taking out food sometimes does not really end up in fasting as they eat for some hours the crumbs.

·        You recommend that the cube could be given in the homecage, but this is not validated by data. As it would be also my gut feeling to recommend that, perhaps you find some literature to include some sentences in the discussion to talk about that.

·        And second for the recommendation: If done in homecage, any idea to avoid that the bedding is sticking to the cubicle? We did not really find any practical solution up to now. Plastic is dangerous and metal bowl are also increasing neophobia…

Author Response

Dear Authors,

thanks for the nice study and the nice idea to refine oral gavage in rats. Oral gavage is stress for rats and humans and a method to circumvent this situation is highly appreciated from my side! So, a method to use voluntary eating would be perfect. Unfortunately, not all animals respond as they should, which is always a problem using voluntary appetitive behavior. I have one main question and some points to clarify, so this would be a minor revision.

My main question is: The results part seems to be three experiments, one with the question “response to training in the focus of age and sex”, the second with the question of the flavor and third the lactating issue. First and third, the animal numbers are clear, but the second experiment, I get confused. Are the animals used again (so they already had undergone the training from part one)? That would mean they are not naïve anymore which would perhaps give an idea why they are more “non-habituating” to the flavors? If this is a new group, then the animal numbers in the method part are not fitting. Perhaps you can clarify my confusion.

ANSWER. All rats were naïve to the stimulus. We made a mistake in the final number of rats, since the fasted rats were not included in the total number. We have amended this and also included 6 additional male rats (fasted rats) and 2 additional lactating rats.

Then some less important questions:

  • Have the rats/animal housing facility SPF status.

ANSWER. Our Animal House Facility does not have the SPF status. However, the rats are regularly monitored regarding sanitary control with very good results.

  • Did you change the cage for fasting the rats? As rats eat when fed ad lib. not very “clean” but rather with a lot of crumbles, taking out food sometimes does not really end up in fasting as they eat for some hours the crumbs.

ANSWER: This is a good point to consider. The rats were kept in their own cage with the bedding to avoid discomfort and we only removed the food from hopper. Therefore, we must consider the possibility that they eat some of the food remaining inside the cage, but there will be minimal.

  • You recommend that the cube could be given in the homecage, but this is not validated by data. As it would be also my gut feeling to recommend that, perhaps you find some literature to include some sentences in the discussion to talk about that.

ANSWER. We also think that supplementation in the own cage would be more comfortable, and we have evaluated if the neophobic males could accept the gelatin in their own cage. The results were positive and we have now included this aspect in “Limitations and protocol implementation”.

  • And second for the recommendation: If done in homecage, any idea to avoid that the bedding is sticking to the cubicle? We did not really find any practical solution up to now. Plastic is dangerous and metal bowl are also increasing neophobia…

ANSWER. We have tried putting the cube in the hopper where they usually eat and with positive results. However, we propose to use this only in the initial raining and then place the rats in the empty cage to ensure dosage of the supplement.

Reviewer 3 Report

The authors assessed the feasibility of using gelatin cubes for oral delivery of drugs to rats as a refinement compared to oral gavage. While some, especially adult male rats, did not habituate to eating the cubes, most animals could be trained to ingest the cubes within minutes and the training was remembered 2 months later. No side effects were noted, including in treatment of lactating dams.   

The presented work is of high interest to the scientific community and aligns well with the scope of the journal since it is relevant for research in experimental animals and offers a refinement approach.

I recommend the acceptance of the manuscript after addressing a few comments:

Major comment:

The group sizes for comparisons in Fig. 3 (additives) are very different. Regarding the variability (and non-normal distribution) in non-habituating rats across groups from Fig. 1, it is necessary to perform a power analysis to identify the required group sizes and potentially match the numbers across groups.

Minor comments:

Methods/Statistics/Results:

The percentages given in the abstract do not match the values for adult male and female rats for gelatine cubes without additives (Fig. 1). Furthermore, the habituated/non-habituated proportions should be compared using contingency tests, e.g. Fisher’s exact test.

Discussion:

The authors should discuss how their gelatin cube approach compares to drug delivery via drinking water (advantages/disadvantages). Some studies using gelatin cubes for delivery of pain medications in rats have already been published (e.g. Cannon et al. Lab Anim 2010. doi: 10.1038/laban1110-342; Flecknell et al. Lab Anim. 1999. doi: 10.1258/002367799780578381). These showed conflicting results regarding acceptance of the cubes, but also efficacy of the delivered drugs compared to systemic administration. These should be discussed.

Author Response

I recommend the acceptance of the manuscript after addressing a few comments:

Major comment:

The group sizes for comparisons in Fig. 3 (additives) are very different. Regarding the variability (and non-normal distribution) in non-habituating rats across groups from Fig. 1, it is necessary to perform a power analysis to identify the required group sizes and potentially match the numbers across groups.

ANSWER: We have performed a power analysis for an F distribution, considering the following parameters: a probability error = 0.05 and effect size=0.7 (2 units of estimated variance) and we obtained an 89% of statistical power for females and 88% for males.

Minor comments:

Methods/Statistics/Results: The percentages given in the abstract do not match the values for adult male and female rats for gelatine cubes without additives (Fig. 1). Furthermore, the habituated/non-habituated proportions should be compared using contingency tests, e.g. Fisher’s exact test.

ANSWER:  We are sorry for the mistake in the abstract, and we have now amended it. We appreciate your suggestion and we have now performed Fisher´s exact test to analyze the proportion of non-habituated animals for figure 1A and figure 3A.

Discussion: The authors should discuss how their gelatin cube approach compares to drug delivery via drinking water (advantages/disadvantages). Some studies using gelatin cubes for delivery of pain medications in rats have already been published (e.g. Cannon et al. Lab Anim 2010. doi: 10.1038/laban1110-342; Flecknell et al. Lab Anim. 1999. doi: 10.1258/002367799780578381). These showed conflicting results regarding acceptance of the cubes, but also efficacy of the delivered drugs compared to systemic administration. These should be discussed.

ANSWER: This is an interesting point which deserves discussion. The lack of effectiveness of oral route in the study of Cannon and co-workers may be related to the metabolism of the drug, particularly the fact that it was administered acutely (60 min before the test). Therefore, it is possible that oral route may not be a good method for acute administration of drugs. However, it should be used to test the effectiveness of foods and possible nutraceuticals, as in the present study, which need to pass through the gastrointestinal digestion to achieve effects. We have demonstrated in a preliminary metabolomic study (still ongoing) that caffeine and theobromine,  bioactive compounds from CSE used in the present study in lactating rats are present in plasma after supplementation. This has now been included in the text (lines 397-400). Regarding the issue of drinking water versus gelatin, we think the drinking water would yield similar results (i.e. influence of drug metabolism and concentrations reaching plasma), since the critical point may be the passage through the digestive tube. Besides, delivery in drinking water presents the disadvantage of the amount of water and drug that the rat consumes. These aspects have been incorporated in the discussion (lines 436-443).

Reviewer 4 Report

The article explores a non-invasive method for oral administration, analyzing the effect of age, sex and gelatin presentation on the latency time for consumption in Sprague Dawley rats. Refining administration methods is an important area in laboratory animal sciences. The article provides potentially interesting results but the design of the experiment could be improved as well as the description of the information:

-       English revision is required

-       Detailed information is required for colony maintenance and genetic control if the strain in produced internally; specific number of animals per cage used for females, males and according to ages; details on the microbiological status of animals.

-       A two hours repeated isolation period seems a long period and may result in additional stress to the animal. This is not discussed by the authors neither compared with potentially more or less stressful methods and should not be ignored. It is not clear if the cumulative effect of repeated isolation 2h periods is higher than the effect of the alternative method (oral gavage without isolation and without fasting). Repeated short isolation periods have been described as having effects on the animals (example: https://pubmed.ncbi.nlm.nih.gov/25510393/).

-       The fasting period was determined as 4 hours but no information is given on what time of the light cycle it was performed and this may also have an effect on animal welfare as these are nocturnal animals.

Author Response

ANSWER: Thank you for your time to revise our manuscript.

  • English revision is required.

ANSWER: We have revised the text, and we hope the English language is correct.

  • Detailed information is required for colony maintenance and genetic control if the strain in produced internally; specific number of animals per cage used for females, males and according to ages; details on the microbiological status of animals.

ANSWER: Our colony stock of Sprague Dawley was acquired in 2019 and last year we introduced new females, to maintain a stock. Our Animal House facility performs regular microbiological analysis to ensure the rats are devoid of pathogens interfering with wellbeing and experiments. Adult rats, both males and females, over 300 gr are kept in groups of 3 and young rats are housed in groups of 6, always in groups of the same sex from weaning.

  • A two hours repeated isolation period seems a long period and may result in additional stress to the animal. This is not discussed by the authors neither compared with potentially more or less stressful methods and should not be ignored. It is not clear if the cumulative effect of repeated isolation 2h periods is higher than the effect of the alternative method (oral gavage without isolation and without fasting). Repeated short isolation periods have been described as having effects on the animals (example: https://pubmed.ncbi.nlm.nih.gov/25510393/).

ANSWER: We agree that isolation can be a stress factor. Thank you for the reference regarding the effect of isolation on stress response. Our data show that adult males seem to be more affected by any stress factor, and isolation may be one of them. We have now included additional experiments presenting the gelatin in the cage of the animals to reduce stress. With this protocol, we evidenced that non-habituated males improved the acceptance. This aspect is now discussed in the last new section where we also propose an implemented training method, using this approach. It must be noted that for supplementation, isolation would be necessary, but it is not likely to interfere with wellbeing and cause stress since it only lasts a few minutes, as we have demonstrated.

  • The fasting period was determined as 4 hours but no information is given on what time of the light cycle it was performed and this may also have an effect on animal welfare as these are nocturnal animals.

ANSWER: The time of the 4h fasting period was from 7:00 to 11:00h.

Reviewer 5 Report

I suggest the authors receive a chance to revise and resubmit.

The animal use in this project is within ethical standards, and indeed, one of its aims is to improve the ethical treatment of animals.

Most important: Authors need to tell us what new information they are offering. Neophobia is well-studied in rats, and others have suggested that voluntary ingestion of test substances could be an important refinement in toxicity testing.

If your conclusions offer some new recommendations, I’d be sure to include the caveats too: that you might need to pre-screen animals for those who consume the GC (so, a concern about Reduction, if you must start with more animals than you will need for analysis, especially if working with adult males)

Second most important:  Animal numbers.

I am not fully competent to judge the statistical analysis and will leave that for another reviewer. But I am concerned at the low numbers of animals.

How did you determine your sample sizes in advance of running the assays? Did that include plans for testing habituated versus non-habituated as well as the subsequent tests of the habituated animals?

Why so few young females? Why only 5 lactating females?

It may be that this is best considered some pilot work, and that you focus on replicating parts of it:  seems unlikely to be successful, but can you try short-term fasting to increase the habituation rate in adult males? Should you replicate in home cages? Develop the work with lactating females, but with more than 5 (again: if you were to need five for analysis in a tox study, how many would you need to start with to get 5?)

Some comments, line by line:

ll. 62-3   perhaps give a quantitative sense of the frequency of these complications.

l. 68 “intricate” is not the best English word for this   

l. 126   explain why you chose CSE for your test substance

l. 137   good they were all done at the same time of day; I would specify; morning? Afternoon?

l. 226   Surprised that only adult females were tested for the effect of fasting.  Just because their nonfasted behaviors were similar to the other groups does not mean other groups would have responded the same to fasting.

Likewise surprised you did not look at whether fasting would improve the proportion of habituated to non-habituated animals.  Your reference #19 (Zhang 2021) fasted mice overnight to get them to eat their jelly.

l. 246 – for the one-month memory assay, it seems you are down to 6 male and 6 females, for some reason, and only the animals trained as adults?  And then you further drop down to 5 lactating females?

l. 281 - 4   I’m not sure that reference #16 is best for supporting the claim about using gelatin.  Or perhaps, change “…it has been proposed that it can be used to” to “it has been used to…”  Better if you have a reference in which someone compared glucose-in-gelatin to some other way of doing a glucose tolerance test.

l. 304:  I would not zero in on sexual maturity as the difference between younger and older rats; enough to say they are different without conjecture in what way.

l. 346:  Same comment about conjecture of untested reasons --- in this case, previous stress – for some of the neophobia you observed.

ll. 361-2    “We conclude that in rats, food additives or sweeteners do not improve acceptance.”  Maybe change to ““We conclude that the food additives and sweeteners we tested (vanilla, sucralose) do not improve acceptance of gelatin in Sprague Dawley rats.”

Author Response

ANSWER: Thank you for your suggestions and comments. We have taken them into account and included additional data, which we hope improves the comprehension of the manuscript.

The animal use in this project is within ethical standards, and indeed, one of its aims is to improve the ethical treatment of animals.

Most important: Authors need to tell us what new information they are offering. Neophobia is well-studied in rats, and others have suggested that voluntary ingestion of test substances could be an important refinement in toxicity testing.

If your conclusions offer some new recommendations, I’d be sure to include the caveats too: that you might need to pre-screen animals for those who consume the GC (so, a concern about Reduction, if you must start with more animals than you will need for analysis, especially if working with adult males)

ANSWER. It is true that neophobia has been well studied previously, but with a different perspective. Our aim was not to gain insight in this process, but to explore factors affecting voluntary ingestion of supplements to provide key elements to develop a protocol to substitute oral gavage by voluntary ingestion. The novelty of our study is our approach to the study of neophobia from the practical point of view and improving animal welfare, since we provide relevant information for researchers to use this method of supplementation which is less aggressive.

We are aware of the limitations of the study, and we have highlighted this aspect in the new version, in a separate section of the discussion “limitations and protocol implementation”. Regarding the need of a larger number of animals to start with, considering those that will reject the supplement, we think this is not a problem, since these animals can be used as controls without supplementation. This is what we currently do.

Second most important:  Animal numbers.

I am not fully competent to judge the statistical analysis and will leave that for another reviewer. But I am concerned at the low numbers of animals.

How did you determine your sample sizes in advance of running the assays? Did that include plans for testing habituated versus non-habituated as well as the subsequent tests of the habituated animals?

Why so few young females? Why only 5 lactating females?

ANSWER. We did not calculate sample sizes and we used all the animals being used in our current study, as part of our 3R strategy, to reduce rat number. We have now 2 additional lactating rats supplemented with CSE included in the study, which we report in the new version.

It may be that this is best considered some pilot work, and that you focus on replicating parts of it:  seems unlikely to be successful, but can you try short-term fasting to increase the habituation rate in adult males? Should you replicate in home cages? Develop the work with lactating females, but with more than 5 (again: if you were to need five for analysis in a tox study, how many would you need to start with to get 5?)

ANSWER: We do not understand on which grounds you base that we cannot replicate our protocol. As mentioned above, we have now continued with our study and have confirmed the successful training of 2 additional lactating rats (now included in the study). We have included a sentence in discussion (lines 275-277). You made a good point regarding the fasting and we have now included a graph with 6 additional adult male rats. Besides, we have performed a pilot protocol, using neophobic males to improve acceptance through presentation of the gelatin in their own cage.

Some comments, line by line:

  • 62-3 perhaps give a quantitative sense of the frequency of these complications.

ANSWER. We have included as sentence.

  • 68 “intricate” is not the best English word for this

ANSWER. We have substituted this word by “detrimental”.

  • 126 explain why you chose CSE for your test substance

ANSWER. The present work is part of an ongoing study of the effects of cocoa shell, a by-product from the chocolate manufacturing industry, which exhibits vasodilatory properties in vitro (PMID: 35204310). Since this is a food matrix which needs to pass through the digestive system we are currently testing it in vivo through oral administration.  We aimed to evaluate the direct effects as well as the effects through lactation, a particularly critical period during which oral gavage may not be suitable due to stress. We have included this aspect in the introduction (lines 88-92) and in the discussion (lines 397-400).

  • 137 good they were all done at the same time of day; I would specify; morning? Afternoon?

ANSWER. All experiments were performed in the morning (9:00h-12:00h). This has been included in the text.

  • 226 Surprised that only adult females were tested for the effect of fasting. Just because their nonfasted behaviors were similar to the other groups does not mean other groups would have responded the same to fasting.

ANSWER. We have now included adult males with fasting period as suggested (Figure 5).

Likewise surprised you did not look at whether fasting would improve the proportion of habituated to non-habituated animals. Your reference #19 (Zhang 2021) fasted mice overnight to get them to eat their jelly.

ANSWER. We preferred to avoid fasting, particularly overnight, which may modify metabolism and cause additional stress. We have used the short fasting and we now report that some of males which maintained neophobia, eat the cube after a short fasting period.

  • 246 – for the one-month memory assay, it seems you are down to 6 male and 6 females, for some reason, and only the animals trained as adults? And then you further drop down to 5 lactating females?

ANSWER. We have now additional lactating females included, which confirms the memory assays.

  • 281 - 4 I’m not sure that reference #16 is best for supporting the claim about using gelatin. Or perhaps, change “…it has been proposed that it can be used to” to “it has been used to…”  Better if you have a reference in which someone compared glucose-in-gelatin to some other way of doing a glucose tolerance test.

ANSWER. We have changed the reference to that of Zhang et al, 2021 in which this protocol is proposed and modified the sentence.

  • 304: I would not zero in on sexual maturity as the difference between younger and older rats; enough to say they are different without conjecture in what way.

ANSWER. We have modified the sentence to avoid focusing on this speculative issue.

  • 346: Same comment about conjecture of untested reasons --- in this case, previous stress – for some of the neophobia you observed.

ANSWER. We have deleted the sentence, as suggested.

  • 361-2 “We conclude that in rats, food additives or sweeteners do not improve acceptance.” Maybe change to ““We conclude that the food additives and sweeteners we tested (vanilla, sucralose) do not improve acceptance of gelatin in Sprague Dawley rats.”

ANSWER. We agree. Other additives may improve acceptance and also other rodents may respond differently.  We have modified the sentence.

Round 2

Reviewer 1 Report

The manuscript has improved significantly and I am satisfied with the current version

Author Response

Answer: We appreciate your time reviewing our manuscript, which it was improved thanks your suggestions.

Reviewer 3 Report

The authors have addressed my concerns, some remaining typos should be corrected beefore publication, see below:

l. 45: (P=0,501) change to (P=0.501)

l. 51 word missing in added sentence

l. 173 change 4,728 to 4.728

l. 176 change 6,440 to 6.440

l. 195 change place to placed

Author Response

Answer: Thank you for your comments. We have changed the text according to your suggestions. Regarding line 51, we have changed the sentence to: “The current protocol may be implemented by training the rats in their own cage”.

Reviewer 4 Report

All comments were addressed. Information on the fasting period (time of the day) could be at least be included in the methods section.

Author Response

Answer: Thank you for your comments. We have included in materials and methods the time for fasting.

Reviewer 5 Report

The authors have addressed all of my comments.

As mentioned in the first review, I dod not feel competent to evaluate the statistical analysis, so I hope you have soneone who is reviewing that.

One sentence of concern on line 266 in the revision:  “These animals can be used as control experiments, contributing to reduction”   I would reword that to something like “These animals may be useful for other studies that do not require voluntary ingestion of compounds.”  As written, I almost thought you meant using them as controls in the same study that were first enrolled for, which would upend any efforts at randomization.

Thanks to the authors for their efforts at improving rat welfare and for sharing with others.

Author Response

Answer: Thank you for your comment that was implemented in the discussion section. Regarding the statistical analysis, we have already reviewed the manuscript according to the reviewer #3.